# COVID-19-Related Cholangiopathy: Histological Findings

**DOI:** 10.3390/diagnostics14161804

**Published:** 2024-08-19

**Authors:** Valéria F. A. Borges, Helma P. Cotrim, Antônio Ricardo C. F. Andrade, Liliana S. C. Mendes, Francisco G. C. Penna, Marcelo C. Silva, Frederico C. Salomão, Luiz A. R. Freitas

**Affiliations:** 1Postgraduate Program in Medicine and Health, Federal University of Bahia, Salvador 40026-010, Brazil; valeriaborges@ufba.br; 2School of Medicine of Bahia, Federal University of Bahia, Salvador 40026-010, Brazil; arcfa01@gmail.com (A.R.C.F.A.); freitas.luizar@gmail.com (L.A.R.F.); 3Hospital de Base do Distrito Federal, Brasilia 70330-150, Brazil; mendesliliana2@gmail.com; 4Department of Internal Medicine, School of Medicine, Federal University of Minas Gerais, Belo Horizonte 31270-901, Brazil; cancelapenna@gmail.com; 5Hospital e Clínica São Roque, Ipiau 45570-000, Brazil; drmarcelocsilva62@gmail.com; 6Centro Diagnóstico de Patologia, Uberlandia 38400-110, Brazil; fredericosalomao@yahoo.com.br; 7Gonçalo Moniz Institute, Oswaldo Cruz Foundation (FioCruz), Salvador 402596-710, Brazil

**Keywords:** bile duct diseases, pathology, cholestasis, COVID-19, sclerosing cholangitis, cholangiopathy, post-COVID-19 cholangiopathy, secondary sclerosing cholangitis in critically ill patients

## Abstract

Cholangiopathy has been described in survivors of severe COVID-19, presenting significant clinical parallels to the pre-pandemic condition of secondary sclerosing cholangitis in critically ill patients (SSC-CIP). We aimed to examine the liver histopathology of individuals with persistent cholestasis after severe COVID-19. Methods: We subjected post-COVID-19 cholestasis liver samples to routine staining techniques and cytokeratin 7 immunostaining and semi-quantitatively analyzed the portal and parenchymal changes. Results: All ten patients, five men, had a median age of 56, an interquartile range (IQR) of 51–60, and required intensive care unit and mechanical ventilation. The median and IQR liver enzyme concentrations proximal to biopsy were in IU/L: ALP 645 (390–1256); GGT 925 (664–2169); ALT 100 (86–113); AST 87 (68–106); and bilirubin 4 (1–9) mg/dL. Imaging revealed intrahepatic bile duct anomalies and biliary casts. We performed biopsies at a median of 203 (150–249) days after molecular confirmation of infection. We found portal and periportal fibrosis, moderate-to-severe ductular proliferation, and bile duct dystrophy in all patients, while we observed hepatocyte biliary metaplasia in all tested cases. We observed mild-to-severe parenchymal cholestasis and bile plugs in nine and six cases. We also observed mild swelling of the arteriolar endothelial cells in five patients. We observed a thrombus in a small portal vein branch and mild periductal fibrosis in one case each. One patient developed multiple small biliary infarctions. We did not observe ductopenia in any patient. Conclusions: The alterations were like those observed in SSC-CIP; however, pronounced swelling of endothelial cells, necrosis of the vessel walls, and thrombosis in small vessels were notable.

## 1. Introduction

Patients recovering from severe COVID-19 exhibit a quantifiable risk of developing COVID-19-related cholangiopathy. The incidence rate is 0.59% among all hospital admissions, 2.3% among mechanically ventilated patients, and 7.8% among critical care hospitalization survivors [1,2,3]. This condition resembles secondary sclerosing cholangitis in critically ill patients (SSC-CIP), first identified in 2001 (SSC-CIP) [4,5,6].

In January 2021, Roth et al. reported three cases of cholangiopathy in patients recovering from severe COVID-19. Liver biopsies revealed bile duct reduction, cholangiocyte damage, endothelial cell swelling, and portal vein endophlebitis. Additionally, one patient exhibited hepatocyte biliary metaplasia. Immunohistochemistry and in situ hybridization tests for SARS-CoV-2 were consistently negative in all cases. The study observed similarities with SSC-CIP and distinct features of post-COVID-19 cholangiopathy [7]. Faruqui et al. presented a case series in 2020, and subsequently, they published findings on 12 patients with persistent, worsening cholestasis after severe COVID-19, which are indicative of SSC-CIP patterns [1,8]. 

This study aimed to examine the liver tissue of individuals with persistent cholestasis following severe COVID-19 to better understand the histopathology changes in the liver associated with this condition.

## 2. Materials and Methods

Adult patients with persistent post-COVID-19 cholestasis were consecutively enrolled and underwent a liver biopsy as clinically advised. We confirmed SARS-CoV-2 infection through molecular assay using reverse transcription polymerase chain reaction (RT-PCR) testing. All biopsies were performed after resolving the infectious condition, specifically following hospital discharge after SARS-CoV-2 infection. We excluded other etiologies of previous or concurrent cholestatic liver disease. None of the patients had a prior history of biliary tract disease. All other potential causes of chronic liver diseases were excluded during the pre-liver biopsy investigation period. All patients tested negative for antinuclear antibodies (ANA), anti-mitochondrial antibodies (AMA), hepatitis B and C serologies, and had no history of alcohol abuse. Furthermore, an assessment of their medication history revealed no prior drug exposure that could be associated with the cholestasis observed in these patients, except for sedation with ketamine during ICU admission [9,10,11,12,13]. 

A single pathologist with expertise in hepatic pathology (LARF) reviewed all patients. Histological liver sections underwent routine staining for liver analysis: Hematoxylin and eosin (H&E), Sirius red, Periodic acid Schiff reaction (PAS), and PAS-Diastase. Furthermore, we performed immunohistochemistry for cytokeratin 7 to assess ductular reactions indicative of ductular proliferation and biliary metaplasia in hepatocytes. The systematic semiquantitative evaluation included alterations within the portal tract (fibrosis, inflammation, biliary, and vascular alterations) and parenchymal changes (cholestasis, inflammation, and hepatocellular damage). The portal inflammation was evaluated using a semi-quantitative and categorical approach, which included precise classifications such as absent, mild, moderate, or severe. Mild inflammation was characterized by a few infiltrating inflammatory cells in most, but not all, portal spaces. Moderate inflammation involved nearly all portal spaces, with many identifiable inflammatory cells even at lower magnifications. Severe inflammation was defined by an infiltration that filled and enlarged the portal spaces [14]. The swelling of endothelial cells in branches of the hepatic artery within the portal tracts was characterized by a swollen, vacuolated appearance. Our intention was not to quantify this finding but to report its occurrence. The process of necrosis, defined as the sequential morphological changes following the death of cells in a living organism, is characterized by increased cytoplasmic eosinophilia, nuclear pyknosis, karyolysis, or karyorrhexis, followed by the drop-out of necrotic cells [14]. The identification of apoptotic cells was based on well-established criteria in tissues stained with H&E [15,16]. 

Study data were collected and managed using REDCap (Research Electronic Data Capture) tools (REDCap 14.0.27—© 2024 Vanderbilt University) hosted at EBSERH (Empresa Brasileira de Serviços Hospitalares, Brasilia, Brazil) [17]. 

## 3. Results

Following a severe infection caused by the SARS-CoV-2 virus, confirmed through molecular diagnosis, forty-two patients (27 males, 64%) experienced persistent cholestasis. The demographic and clinical characteristics of all forty-two patients are included in Appendix A. Ten out of the forty-two patients underwent a hepatic biopsy; the clinical characteristics are detailed in Table 1.

### 3.1. Histopathological Findings

The patients were analyzed, on average, 196.8 ± 58.2 days after testing positive for SARS-CoV-2 (RT-PCR). The median time from a positive RT-PCR test for SARS-CoV-2 to the biopsy procedure was 203 days, with an interquartile range of 150 to 249 days. The biopsy technique involved seven core needle biopsies and three wedge biopsies. The median count of portal tracts in the samples examined was 25, ranging from 15 to 32, indicating that the sampling was satisfactory. The semiquantitative changes that were analyzed in each of the ten cases of post-COVID-19 cholangiopathy are described in Appendix A. There was evidence of portal, periportal, and septal fibrosis in all cases, but the extent of fibrosis varied. Some cases exhibited marked portal and septal fibrosis, while others showed less severe fibrosis (Figure 1A,C). One patient was diagnosed with biliary cirrhosis. We observed alterations in the bile ducts, a remarkable ductular reaction (Figure 1D), and dystrophy of the biliary ducts and ductules in all patients (Figure 1B,F). We noted vacuolization of cholangiocytes in all but one case. In some cases, this vacuolization was significant; however, in most cases, it was mild (Figure 1E). Additionally, necrotic or apoptotic cholangiocytes were observed (Figure 1B,F). Finally, we frequently observed bile plugs inside the ducts.

The severity of the inflammation in the portal area ranged from mild to moderate. This inflammation primarily included lymphocytes with a few neutrophils and scattered eosinophils (Figure 1B,F). Additionally, inflammatory cells had infiltrated the walls of the ducts and bile ductules (Figure 2A,B). A figure illustrating the severity distribution of inflammatory cells has been included in the Appendix A. The swelling of endothelial cells in the branches of the hepatic artery within the portal tracts was characterized by a swollen, vacuolated appearance. In our series, endothelial swelling was an incidental finding in 50% of cases, specifically in five samples, and a few cases had microthrombi (Figure 2C–E). We observed arteriolar wall necrosis in a few patients (Figure 2F). This aspect is further addressed in the discussion. During the same period in which these ten biopsies were analyzed, we had the opportunity to observe two cases of patients who underwent critical ICU admission and developed cholangiopathy in critically ill patients not related to COVID-19. The histological findings were similar, except for vascular thrombosis and endothelial inflammation, which were exclusive to cases of severe COVID-19-associated cholangiopathy.

The most notable changes in the liver parenchyma were associated with cholestasis. Hepatocytes were soaked in bile, and bile plugs were in the bile canaliculi. Almost all patients exhibited bilirubinostasis, and hepatocyte biliary metaplasia was confirmed by immunostaining for cytokeratin 7. Mild to moderate ballooning of hepatocytes was also observed in all cases. Furthermore, necrotic foci in hepatocytes and scattered isolated apoptotic cells were observed. In one case, there were small areas of biliary infarction (Figure 3A–C).

Table 2 presents a comprehensive overview of the semiquantitative histopathological findings that have been meticulously analyzed, providing a wealth of information for reader understanding and further research.

### 3.2. Follow-Up

During the follow-up period, which lasted up to 1228 days post-COVID-19 diagnosis (median 796 days, IQR 429.2–931.5), 12 of the 42 patients passed away, accounting for almost 30% of this cohort. Among the 12 patients who died during the follow-up period, 11 (91.7%) died due to liver-related causes (decompensation of advanced chronic liver disease) and 1 (8.3%) due to non-liver-related causes (Appendix A).

The biopsy group was followed up for 814 days (with a range of IQR 740.5–882.2) from their SARS-CoV-2 infection diagnosis. During this period, the following clinical observations were noted: two patients died from liver disease, one patient was diagnosed with biliary cirrhosis, five patients developed severe cholestasis, and two participants showed minimal changes in liver tests (maintenance of elevated liver enzymes less than two times the upper limit of normal). The demographic characteristics of the eight surviving patients among the ten who underwent liver biopsy are available in Appendix A.

## 4. Discussion

This study analyzes ten liver biopsy samples collected from patients with COVID-19-related cholangiopathy. Although study was limited in size, this is currently the most extensive published study on the histology of COVID-19-related cholangiopathy in patients without previous biliary liver diseases. In the study by Shih et al., they conducted the most extensive series of seven biopsies to examine post-COVID-19 cholangiopathy to date [18]. The findings of this study are notable as they revealed several features not commonly observed in SSC-CIP. We observed unique microvascular changes in five cases, including swelling of endothelial cells in small branches of the hepatic artery, occasional necrosis of small vessels, and thrombosis of small vessels.

Microvascular alterations, such as microangiopathy, characterized by endothelial swelling with luminal narrowing of hepatic arteries and portal vein endophlebitis, were described in post-COVID-19 cholangiopathy cases [7,19,20]. Esposito et al.’s original work on secondary sclerosing cholangitis in critically ill patients did not achieve this [21]. The mechanism of this alteration was not established in SARS-CoV-2 infection. It may result from a direct cytopathic effect of the virus in the endothelial cells or the morphological translation of endothelial dysfunction associated with microthrombosis. There was no significant correlation among age, sex, or liver enzyme levels and endothelial swelling (Appendix A). There was no significant correlation between the histological findings and the blood coagulation parameters available (Appendix A).

Our histopathological examination of the liver revealed findings consistent with those documented in SSC-CIP, not necessarily related to SARS-CoV-2 infection. These included cholangiocyte injury manifested as degenerative changes (cytoplasmic vacuolization, necrosis, and apoptosis); ductular reaction characterized by a variable extent of ductular proliferation; mild to moderate mixed inflammatory infiltrate within portal tracts; portal and septal fibrosis; concentric periductal fibrosis (two cases); a pattern of biliary cirrhosis (one case); and evidence of bilirubinostasis and hepatocellular cholestasis. Significant cytoplasmic vacuolization was observed in cholangiocytes before and after the onset of the COVID-19 pandemic [7,18,21].

Esposito et al. published a comprehensive publication regarding the histological alterations observed in SSC-CIP, which are like those we have described, except for vascular alterations. The authors suggested that the initial insult occurred in the biliary duct, most likely ischemic [21]. Unlike the hepatic parenchyma, the biliary tree exclusively depends on arterial circulation. The biliary pathway suffers from ischemia and hypoxia, which destroy these structures with secondary alterations [22].

It is of utmost importance to note that COVID-19-related cholangiopathy has been exclusively reported in patients requiring intensive care [2]. An exception to this has been described in only one conference abstract, but in which the patients had preexisting liver diseases [23]. To diagnose SSC-CIP, it is essential to ensure that the patient has no history of biliary liver disease and that there is no known pathological process or injury responsible for blocking the bile duct [24]. The hepatocytes get blood from hepatic arteries and portal vein, while branches of gastroduodenal and hepatic arteries vascularize the common bile duct. The intrahepatic biliary tree only receives its blood supply from the peribiliary plexus, which is highly sensitive to ischemia and blood pressure reduction [22]. In addition to the published studies featuring a substantial number of cases [2,25], our findings support this theory. In this study, all ten patients who underwent a liver biopsy and the entire cohort of forty-two individuals required critical hospitalization in the intensive care unit and were placed on mechanical ventilation. Due to underlying conditions such as severe pneumonia, polytrauma, burns, infections, cardiac surgery, acute respiratory distress syndrome (ARDS), or bleeding after abdominal surgery, patients who developed SSC-CIP required prolonged treatment in the ICU [4,5,26]. COVID-19-related cholangiopathy and SSC-CIP share similar pathophysiological mechanisms. These conditions are believed to be caused by reduced blood flow to the liver and bile ducts, which can result from severe lung injury, shock, and mechanical ventilation [4,5,26]. Furthermore, coagulation system activation and hyperfibrinogenemia can all contribute to sustained systemic hypoxia. These factors can lead to microthrombosis in the intrahepatic branches of the hepatic artery, resulting in disruptions to microvascular blood flow, decreased perfusion, endothelial cell damage, and an increased risk of thrombosis [7,27]. We investigated a series of patients who succumbed to COVID-19 and underwent a minimally invasive autopsy (Freitas LAR, unpublished data). Upon detailed examination of the cases, we observed small, recent thrombi in the hepatic sinusoids of a patient who expired during the acute phase of COVID-19. Some cholangiocytes were vacuolated. This recurring feature suggests that these patients, in some manner during the acute stage of the illness, already exhibited signs of assault on their biliary trees, likely due to ischemia or hypoxia. A publication from our group observed that these patients showed signs of intravascular coagulation despite being subjected to an intensive anticoagulation regimen during pulmonary examination. We observed fibrin within the microcirculation. This observation may be correlated with the hyperfibrinogenemic characteristics recently identified in these patients [28].

Despite the recognized role of the inflammatory cascade as a contributing factor in developing COVID-associated cholangiopathy, Leonhardt et al. did not find differences in classical inflammation parameters between mechanically ventilated patients who developed SSC-CIP and those who did not. Both groups had comparable initial inflammation levels upon admission, as indicated by measurements of C-reactive protein, procalcitonin, interleukin-6, and ferritin [2]. In our study, there was no control group to compare inflammatory markers. While unexpected, this finding engages us in further exploring the role of inflammation in this condition.

The toxicity of ketamine may also be linked to altered bile composition [9,10,11,13]. Recent evidence has confirmed that ketamine metabolites accumulate in biliary casts [12], and it has been administered in all cases where medication data were available in this cohort, often at doses exceeding customary amounts. Ketamine abuse has been associated with secondary sclerosing cholangitis, regardless of hospitalization or intensive care unit [29].

Although there is a hypothesis that SARS-CoV-2 may have a direct cytopathic effect on cholangiocytes due to a high concentration of ACE-2 receptors, no definitive evidence supports this claim. No studies have yet shown the presence of SARS-CoV-2 within biliary cells in cases of post-COVID cholangiopathy [30,31,32]. One limitation of our study is that we did not employ methods to detect SARS-CoV-2 in liver tissue specimens. However, studies suggest SARS-CoV-2 could contribute to biliary pathology through microangiopathy, characterized by endothelial cell damage and thrombotic events [7,18,27].

We posit that the primary target is the hepatic arterial microvasculature, which consequentially affects the intrahepatic biliary tree and cholangiocytes—cells entirely dependent on this vasculature. In addition, thrombotic phenomena and compromised flow within the hepatic arterial network can primarily lead to biliary duct injury.

COVID-19-related cholangiopathy presents as a spectrum disorder, with certain patients experiencing a positive response over time, while others may suffer a progression of the condition that may ultimately result in liver failure, transplantation, and mortality [33,34,35,36].

Many studies have involved patients who initially reported the typical findings of secondary sclerosing cholangitis [37]. However, not all our patients showed radiological changes consistent with this condition, and they did not have stenosis or dilation of the biliary tract. Nonetheless, they exhibited cholestasis and underwent a biopsy, which confirmed the presence of biliary disease. The destruction of the bile duct wall stood out more than the development of well-defined strictures [38] and the extrahepatic biliary tree was usually spared [37,38]. The evolution of COVID-19-related cholangiopathy and SSC-CIP to secondary biliary cirrhosis is faster than in other secondary sclerosing cholangiopathies [39,40,41]. 

This study has some limitations. These include a small number of cases, the absence of large portal tracts in the samples, and the need for methods to detect SARS-CoV-2 in the tissue. Another limitation is the lack of biopsies conducted at two different time points. We acknowledge that this is a regrettable limitation. The pathological findings observed during the peak of SARS-CoV-2 infection may or may not have included patients who would eventually develop COVID-19-associated cholangiopathy [32]. Notably, Roth et al. identified this condition as post-COVID-19 cholangiopathy in their study “Post-COVID-19 Cholangiopathy: A Novel Entity” [7]. Liver biopsies performed on patients while they were receiving care for COVID-19 and then again when the chronic cholestasis condition was established would have added significant value to the studies.

## 5. Conclusions

Overall, the cases of COVID-19-related cholangiopathy analyzed herein presented histological alterations like those observed in secondary sclerosing cholangitis in critically ill patients (SSC-CIP), patients. Notably, the pronounced swelling of endothelial cells, necrosis of the vessel walls, and thrombosis in small vessels are novel findings, only described in post-COVID-19 cholangiopathy, that require further attention.

## Figures and Tables

**Figure 1 diagnostics-14-01804-f001:**
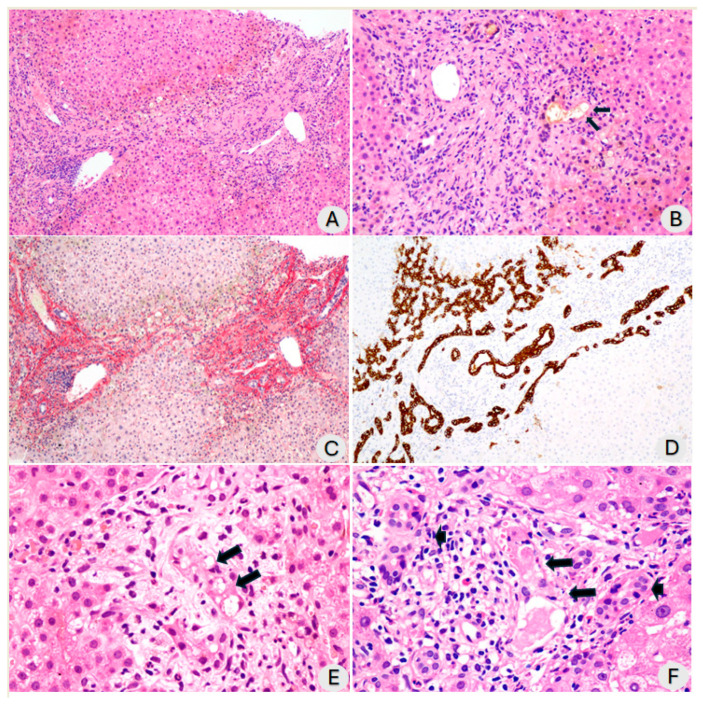
(**A**) Fibrous widening of interconnected portal tracts. Discrete to moderate inflammatory infiltration of mononuclear cells and ductular reaction was observed (H&E, ×100); (**B**) Fibrous widening of portal spaces with ductular reaction and discrete inflammatory infiltration of mononuclear cells. Dystrophic bile duct and a ductule with necrosis of cholangiocytes containing bile in the lumen can be observed (arrows) (H&E, ×200); (**C**) Same area as shown in (**A**), stained with Sirius red to demonstrate portal fibrosis. The ductular reaction can be observed more easily (Sirius red, ×100); (**D**) Immunohistochemical staining with anti-cytokeratin 7 antibody highlights intense ductular reaction and dystrophic bile duct in the center of the portal tract (×100); (**E**) Portal tract containing a bile duct with evident vacuolization of cholangiocyte cytoplasm (arrows) (H&E, ×200); (**F**) Portal tract exhibiting altered bile ducts. Long arrows indicate ducts with necrosis of cholangiocytes. Short arrows indicate dystrophic bile ducts. Discrete inflammatory infiltration of mononuclear cells and some eosinophils can be observed (H&E, ×200).

**Figure 2 diagnostics-14-01804-f002:**
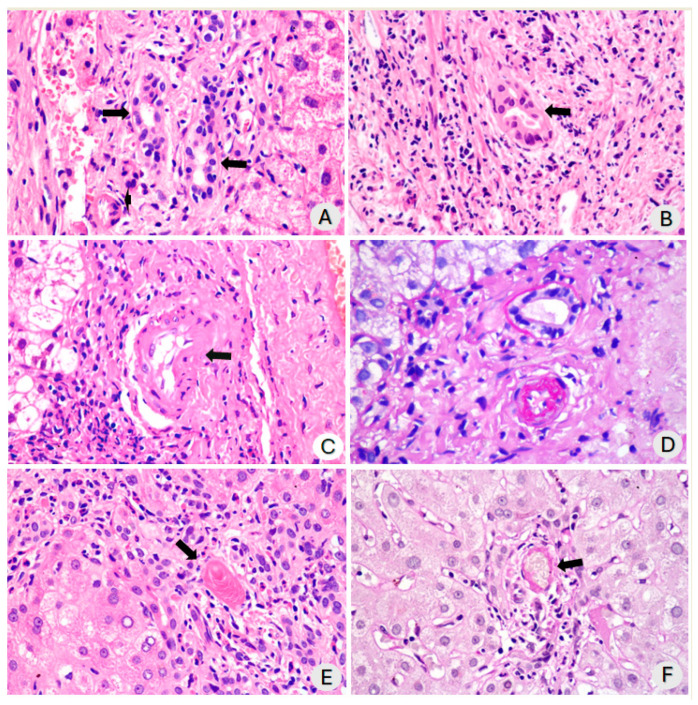
(**A**) Portal tract showing dystrophic bile ducts with infiltration of some lymphocytes in the wall (arrows) (H&E, ×200); (**B**) Expansion of the portal tract due to fibrosis with slight mononuclear inflammatory infiltration. The arrow indicates dystrophic ducts with some lymphocytes in the wall (arrow) (H&E, ×200); (**C**) The arrow indicates an artery with vacuolization of endothelial cell cytoplasm (H&E, ×200); (**D**) An arteriole can be observed with vacuolization of endothelial cells and infiltration of the wall by PAS+ material, resistant to diastase. Above the arteriole, there is a dystrophic bile ductule with irregularly arranged cholangiocytes, hyperchromatic nuclei, and slight vacuolization of the cytoplasm can be observed (PAS with diastase, ×200); (**E**) Portal tract with slight mononuclear inflammatory infiltration, ductular reaction, and in the center (arrow) a blood vessel occluded by a recent fibrinous thrombus (H&E, ×200); (**F**) Small portal tract, exhibiting a vessel with wall necrosis. The arrow points to a necrotic vessel with two cells displaying pyknotic nuclei and increased cytoplasmic eosinophilia (H&E, ×200).

**Figure 3 diagnostics-14-01804-f003:**
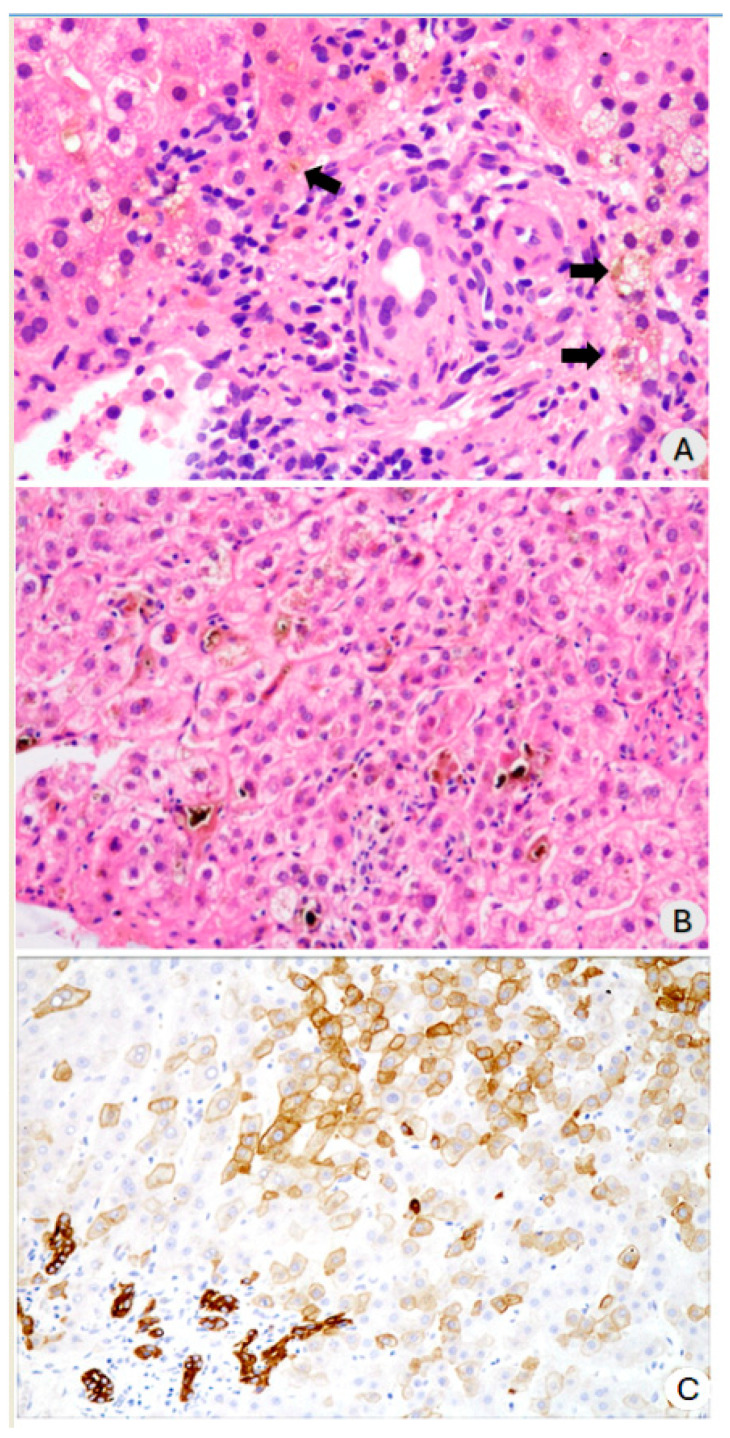
(**A**) The portal tract with dystrophic duct and mild mononuclear inflammatory infiltration. The arrows indicate hepatocytes in the periportal region with cytoplasmic bile impregnation, some slightly ballooned (H&E, ×200); (**B**) The centrolobular area of the hepatic parenchyma (zone 3) and zone 2 showing biliary thrombi in the bile canaliculi and hepatocytes, with wider and clearer cytoplasm. A focus of necrosis of isolated hepatocytes with inflammatory cell infiltration can be observed (H&E, ×200); (**C**) Immunostaining with anti-cytokeratin 7 antibody, showing more intense staining of the bile ducts. The lighter staining in the parenchyma corresponds to hepatocytes with biliary metaplasia (×100).

**Table 1 diagnostics-14-01804-t001:** Clinical characteristics of ten patients who underwent a liver biopsy.

Gender, male, *n* (%)	5	(50%)
Molecular diagnosis of COVID-19, *n* (%)	10	(100%)
ICU admission, *n* (%)	10	(100%)
Mechanical ventilation, *n* (%)	10	(100%)
Age, years		
Median [IQR]	56	[51.2–60]
Mean ± SD	54.4	±12.2
Hospital stays, days		
Median [IQR]	55	[38–60.5]
Mean ± SD	62.1	±34.1
Alkaline phosphatase, U/L		
Median [IQR]	645.5	[389.5–1256]
Mean ± SD	804.8	±593.8
Gamma-glutamyl transferase, U/L		
Median [IQR]	925.2	[663.7–2169]
Mean ± SD	1389	±1094.8
Total bilirubin, mg/dL	3	1–6
Median [IQR]	3.6	[0.9–9.4]
Mean ± SD	6.2	±6.6
Aspartate aminotransferase, U/L		
Median [IQR]	86.8	[67.6–105.7]
Mean ± SD	86.8	±28.1
Alanine aminotransferase, U/L		
Median [IQR]	100.3	[86.2–113.2]
Mean ± SD	107.6	±42.4

IQR: interquartile range; SD: standard deviation; U/L: units per liter; mg/dL: milligrams per deciliter.

**Table 2 diagnostics-14-01804-t002:** Histopathological findings in liver biopsy specimens from ten cases of post-COVID-19 cholangiopathy.

	Median or *n*	Range or %
Number of portal tracts	25	15–32
Portal and periportal fibrosis	10/10	100%
Septal fibrosis	7/10	70%
Periductal fibrosis (mild)	2/10	2%
Inflammatory infiltrate (mild to moderate)	10/10	100%
Bile duct dystrophy	10/10	100%
Bile plugs in bile ductules	6/10	60%
Swelling of endothelial cells of arterioles	5/10	50%
Microthrombi in small vessels	1/10	10%
Ductopenia	0/10	0%
Parenchymal cholestasis (Mild to severe)	9/10	90%
Biliary infarcts (Multiple small)	1/10	10%
Hepatocyte biliary metaplasia (Cytokeratin 7+ cells)	7/7	100%

## Data Availability

The corresponding author, H.P.C., can provide data supporting the study’s findings on request.

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
