# Peer review of "COVID-19-Related Cholangiopathy: Histological Findings"

_diagnostics, 2024, doi:10.3390/diagnostics14161804_

Round 1

Reviewer 1 Report (Previous Reviewer 3)

Comments and Suggestions for Authors

This is an interesting paper. However, the liver biopsy was performed after the COVID-19 infection had improved, and the liver damage may have improved from the values ​​presented, so the pathological findings presented seem to deviate from those seen during the peak of COVID-19 infection.

Author Response

Reviewer 2 Report (New Reviewer)

Comments and Suggestions for Authors

Borges et al report histological findings showing evident biliary inflammatory, fibrotic and ductular reactions in COVID 19 patients admitted in critical care units.

The histological findings are clear and conclusions are supported by these.

I would like to point out minor concerns 

Authors highlighted their histological findings, especially artery endothelial cells swelling, can they speculate on why previous reports did not find these?

Authors should re-structure several sentences to enhance the formality of the manuscript. For instance:

Lines 88, 244, 255, 288

Line 258 Please include a reference supporting the pathological effect of IL-6 on hepatobiliary cells

Comments on the Quality of English Language

This study is concise with convincing evidence and adds to a highly relevant topic, which is post-COVID sequelae. Also, this study is appropriate for the journal. 

Author Response

Reviewer 3 Report (New Reviewer)

Comments and Suggestions for Authors

Nice work on liver histology post Covid changes.  Exhaustive presentation of histological findings. I think that it will be of interest to present more in detail the criteria you used to  assess the severity of the inflammation in the portal area ranged from mild to moderate. No other comments for the rest.

Author Response

Reviewer 4 Report (New Reviewer)

Comments and Suggestions for Authors

I read with immense interest the original manuscript reporting on covid-19 cholangiopathy with the histology features primarily due to hepatic mircocirculatory disturbance, microthrombi, and distinct from usual secondary sclerosing cholangitis of critical illness aetiology. 

I have NO extra inputs and to me the manuscript is well drafted, has relevant discussions within it, authors have covered the topic and its other considerations well enough. 

Round 2

Reviewer 1 Report (Previous Reviewer 3)

Comments and Suggestions for Authors

No additional comments.

Reviewer 2 Report (New Reviewer)

Comments and Suggestions for Authors

The authors have properly improved the manuscript. I find satisfactory the response made by the authors over the suggestions I pointed out.

This manuscript is a resubmission of an earlier submission. The following is a list of the peer review reports and author responses from that submission.

Round 1

Reviewer 1 Report

Comments and Suggestions for Authors

The manuscript is well written but statements are not supported by data. Additional data needs to be provided to support the authors' claims. Please see my comments below-

1. How many days after severe Covid-19 were these patients analyzed? Bile duct damage is an early event in covid-19 infection. Are the authors sure that none of the patients had a previous history of biliary tract disease? How was this confirmed?

These details must be included. 

2. Please provide a supplemental table listing demographic and clinical characteristics of all forty-two patients analyzed. 

3. Images from a control/non-COVID infected liver/patient must be included in all figures.

4. Line 106: Inflammation included infiltration of lymphocytes, few neutrophils and scattered eosinophils. While H&E images can help in identifying subsets of immune cells based off morphology, it is not confirmative. The authors should provide additional histochemistry images to confirm the identity of these immune cells before making such statements.

5. Line 108: Half of the samples showed swelling - what population of patients did this half comprise? Was it mostly males or females, mostly senior citizens? These details need to be included. How do the authors quantify swelling (compared to what)? What is the measure?

6. Line 109: We observed arteriolar wall necrosis? How are the authors identifying necrosis? What markers of necrosis have been checked for?

7. Line 127: How many hepatocytes show ballooning? How many show apoptosis? Is this consistently observed for all patients? How many of these are identified per field? 

Identification of apoptotic cells must be supplemented with an apoptotic marker staining. 

8. Line 133, Figure 3B - It appears that some of these cells are not Zone 3, but also Zone 2. The authors must be careful in data interpretation and confirm their claims by staining with markers of different hepatocyte zones. 

9. Table 2 headings need proper formatting. 

10. Line 148, 149: The authors need to provide data to support their claims for the follow-up study. What were the ages of the patients surviving and available for follow up studies?  

Reviewer 2 Report

Comments and Suggestions for Authors

The authors provide an comprehensive histopathological documentation in a series of 10 patients that developed COVID-19 related cholangiopathy. The number of cases included in the study is limited (only 10), this is the main drawback of the study. however, the level of novelty is high, the histopathological findings are minutiously described and the information provided are valuable for understanding of this pathology.

As minor suggestions:

Table 2 is difficult to read. Please revise it.

Line 144: Why do these patients pass away? can you give more details?

Discussions: Can you discuss if any kind of cholangiopathy and liver fibrosis were encountered in other non- COVID ICU hospitalized patients, with mechanical ventilation?

Reviewer 3 Report

Comments and Suggestions for Authors

This is an interesting paper. Can you describe your liver function before COVID-19 infection? Also, what about the liver tissue after the infection has subsided? Furthermore, we recommend examining the relationship between blood coagulation factors and liver tissue during infection.

Comments on the Quality of English Language

Minor editing of English language required
